# Quantum decoherence of dark pulses in optical microresonators

Chenghao Lao[1,6], Xing Jin[1,6], Lin Chang[2,6], Heming Wang [3], Zhe Lv[1], Weiqiang Xie[3,5], Haowen Shu[2], Xingjun Wang [2], John E. Bowers [3] ✉ & Qi-Fan Yang[1,4] ✉

Quantum fluctuations disrupt the cyclic motions of dissipative Kerr solitons (DKSs) in nonlinear optical microresonators and consequently cause timing jitter of the emitted pulse trains. This problem is translated to the performance of several applications that employ DKSs as compact frequency comb sources. Recently, device manufacturing and noise reduction technologies have advanced to unveil the quantum properties of DKSs. Here we investigate the quantum decoherence of DKSs existing in normal-dispersion microresonators known as dark pulses. By virtue of the very large material nonlinearity, we directly observe the quantum decoherence of dark pulses in an AlGaAs-on-insulator microresonator, and the underlying dynamical processes are resolved by injecting stochastic photons into the microresonators. Moreover, phase correlation measurements show that the uniformity of comb spacing of quantum-limited dark pulses is better than $1.2 \times 10^{-16}$ and $2.5 \times 10^{-13}$ when normalized to the optical carrier frequencies and repetition frequencies, respectively. Comparing DKSs generated in different material platforms explicitly confirms the advantages of dark pulses over bright solitons in terms of quantum-limited coherence. Our work establishes a critical performance assessment of DKSs, providing guidelines for coherence engineering of chip-scale optical frequency combs.

Among the many sources causing decoherence of waves in open systems, quantum fluctuations arising from the perpetual energy exchange with the surroundings are inevitable[1]. Quantum decoherence is a major obstacle for quantum information processing and it also sets a barrier in the classical world, famously exemplified by the Schawlow-Townes linewidth of lasers[2]. This fundamental limit is becoming more important since the precision of laser-based metrology is approaching 21 digits[3].

It is well known that the decoherence in linear systems can be quantified by the fluctuation-dissipation theorem[4], but is complicated by the multipartite interaction in nonlinear systems. As multi-mode optical parametric oscillators, coherently-driven Kerr microresonators provide a fertile ground for investigating nonlinear quantum dynamics[5–7]. In these devices, an array of longitudinal modes are coupled through four-wave mixing – a process seeded by quantum fluctuations (Fig. 1a). With sufficient pump power, multi-mode lasing

[1]State Key Laboratory for Artificial Microstructure and Mesoscopic Physics and Frontiers Science Center for Nano-optoelectronics, School of Physics, Peking University, 100871 Beijing, China. [2]State Key Laboratory of Advanced Optical Communications System and Networks, School of Electronics, Peking University, 100871 Beijing, China. [3]Department of Electrical and Computer Engineering, University of California, Santa Barbara, CA 93106, USA. [4]Collaborative Innovation Center of Extreme Optics, Shanxi University, Taiyuan 030006, China. [5]Present address: State Key Lab of Advanced Optical Communication Systems and Networks, Department of Electronic Engineering, Shanghai Jiao Tong University, Shanghai 200240, China. [6]These authors contributed equally: Chenghao Lao, Xing Jin, Lin Chang. ✉e-mail: bowers@ece.ucsb.edu; leonardoyoung@pku.edu.cn

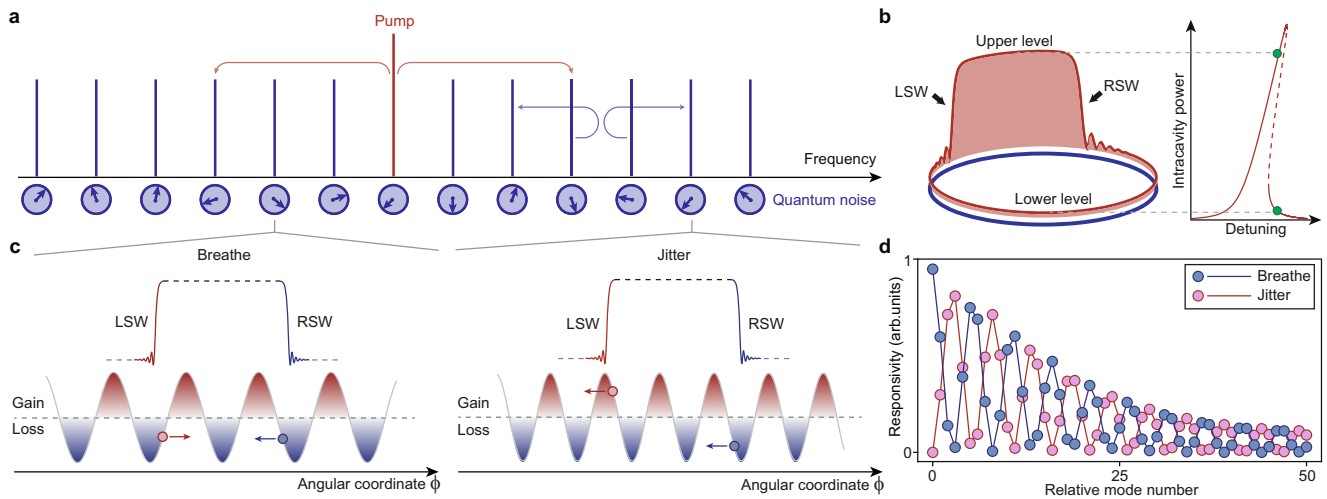

**Fig. 1 | Quantum dynamics of dark pulses. a** Parametric processes in a coherently-pumped Kerr microresonator. Each longitudinal mode is coupled to vacuum electromagnetic fields with random amplitudes and phases. **b** Intensity profile of a typical dark pulse (left panel). The upper and lower levels correspond to the bistable continuous-wave states in the Kerr microresonator (right panel) and are connected by a pair of switching waves (SWs). LSW left SW, RSW right SW. **c** Mechanism of quantum-fluctuation-induced motions of dark pulses. The cases of stochastic photons coupled to two different modes are presented. Regimes featuring net gain and loss are indicated by red and blue shadings, respectively. The arrows show the SWs' moving directions. **d** Simulated responsivity of a dark pulse's motion to quantum fluctuations exerted on a certain longitudinal mode.

takes place, giving rise to various waveforms including Turing patterns, modulation instabilities, and dissipative Kerr solitons (DKSs)[8]. In particular, DKSs are self-organized wavepackets that are manifested as bright solitons[9–15] or dark pulses[10,16–22] depending on the group velocity dispersion (GVD) of the microresonator, and the modes are synchronized to form comb structures in the frequency domain accordingly. The realization of DKSs in microresonators has enabled coherent photonic circuitry for metrology[23–25] and massively parallel data processing[26–28]. Quantum decoherence of DKSs is receiving considerable recent interest as it causes timing jitter of the emitted pulse trains (also referred to as quantum timing jitter) and degrades the mutual coherence of the comb lines, imposing crucial performance limits to their potential applications. Such limits have hitherto been predicted[29] and observed in bright solitons[30–32]. However, even quantum-limited bright solitons have not yet been close to the coherence of bench-top instruments[33]. In this work, we seek other waveforms in microresonators with improved coherence, especially dark pulses which feature compelling power efficiency and spectral flatness for communication and microwave applications[27,28].

## Results

Unlike bright solitons, so far the quantum decoherence process of dark pulses remains unexplored. We first present the theoretical basis describing the quantum dynamics of DKSs in microresonators. The evolution of DKSs obeys the Lugiato–Lefever equation[29,30,34] (LLE), expressed in its dimensionless form as

$$\frac{\partial \psi}{\partial \tau} = id_2\frac{\partial^2 \psi}{\partial \phi^2} + i|\psi|^2\psi - (1+i\zeta)\psi + f + \epsilon(\phi,\tau). \quad (1)$$

The slowly-varying envelope $\psi(\tau,\phi)$ at time $\tau$ and angular position $\phi$ is studied in the frame that rotates around the microresonator at the rate equal to the free spectral range (FSR). The normalized dispersion ($d_2$), detuning ($\zeta$), and pump ($f$) are defined in Methods. Quantum fluctuations are introduced as Langevin force $\epsilon$, which satisfies

$$\langle\epsilon(\phi,\tau)\epsilon^*(\phi',\tau')\rangle = \frac{2\hbar\omega_o^2 n_2 D_1}{\kappa n_o A_{\text{eff}}}\delta(\phi-\phi')\delta(\tau-\tau'), \quad (2)$$

with $\omega_o$ the frequency of the pump, $n_o(n_2)$ the linear (nonlinear) refractive index, $D_1$ the FSR, $\kappa$ the decay rate of the microresonator and $A_{\text{eff}}$ the effective mode area.

At normal GVD, the LLE features two stable continuous-wave solutions at certain pumping conditions (Fig. 1b). The coexistence of the two disparate levels in the microresonator is accompanied by localized wavefronts known as the switching waves (SWs), which appear in pairs to form dark pulses along with the lower level[21,35,36]. Note that the flat-top pulse comprising the upper level and the SWs are often referred to as platicons[16,22]. At equilibrium, the relative position between the SWs (or equivalently the duty cycle defined as the portion occupied by the upper level) is sustained due to the balance between gain and loss. By launching more pump power but keeping other parameters fixed, part of the lower level will be converted to the upper level such that the duty cycle increases[21,35]. Now we consider a stochastic force with amplitude $\epsilon_n$ and phase $\theta_n$ exerted on the $n_{\text{th}}$ mode relative to the pump, and the resulting effective pump term yields

$$|f_{\text{eff}}(\phi,\tau)| \approx f + \epsilon_n\cos(n\phi + \theta_n). \quad (3)$$

The modulated pump divides the microresonator into regimes of net gain or loss compared with the unperturbed situation, as shown in Fig. 1c. If an SW is located in the gain regime, its upper level tends to expand, so that the left SW moves to the left and the right SW moves to the right. Such motions are reversed in the loss regimes. The motions of individual SWs would interfere to cause aggregate motions of the dark pulse. Two situations are thus expected: (1) when both SWs are located in the same regimes, they move in different directions so that the duty cycle breathes and the pulse energy fluctuates; (2) when they are located in opposite regimes, they move in the same direction, causing timing jitter of the dark pulse. The responsivity of the two types of motions to the noise applied on the $n_{\text{th}}$ mode is given by

$$\chi_{\text{jitter}} \propto 1 - \cos(2\pi n\Lambda), \quad (4)$$

$$\chi_{\text{breathe}} \propto 1 + \cos(2\pi n\Lambda), \quad (5)$$

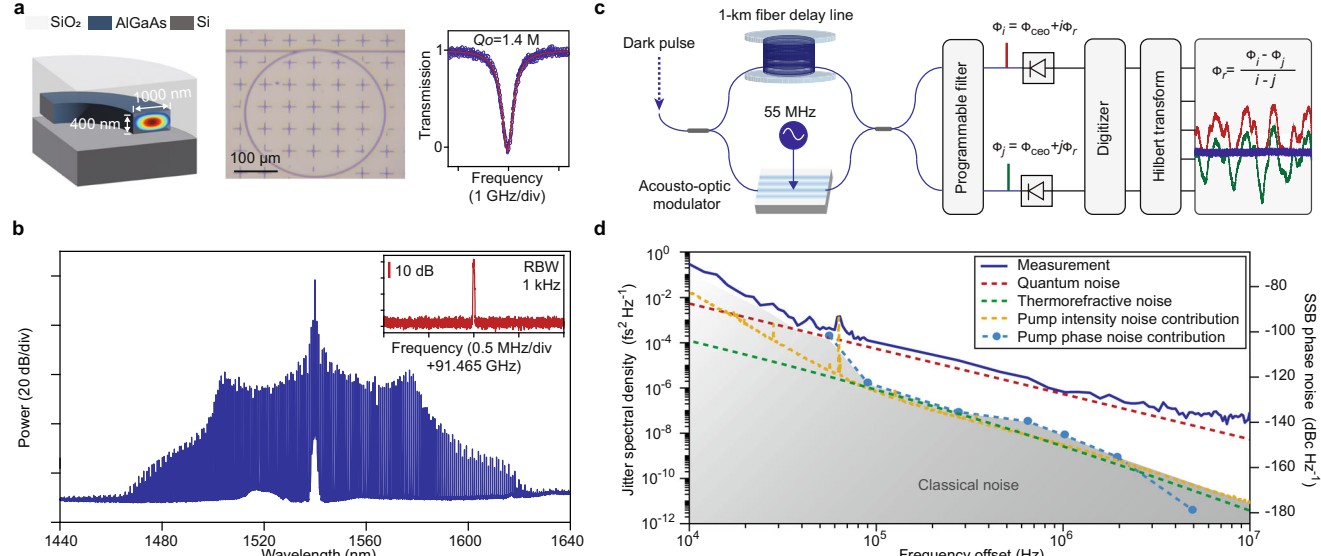

**Fig. 2 | Characterization of dark pulses in AlGaAsOI microresonators. a** Left panel: the cross-sectional view of the microresonator with core dimensions of 400 nm × 1000 nm. The profile of the fundamental TE mode is overlaid. Middle panel: photograph of the AlGaAsOI microresonator used in this study. Right panel: transmission spectrum showing the intrinsic quality factor as 1.4 million. **b** Typical optical spectrum of a dark pulse. Inset: electrical spectrum showing dark pulse repetition frequency. **c** Schematic illustration of the timing jitter measurement. The instantaneous phase of the $i(j)_{th}$ comb line equals $\Phi_{ceo} + i(j)\Phi_r$, with $\Phi_{ceo}$ and $\Phi_r$ associated with the carrier-envelope-offset frequency and repetition frequency, respectively. **d** Measured single-sideband (SSB) jitter spectral density of the dark pulse (blue) and simulated quantum limit (red). Contributions from the intensity (yellow) and phase (cyan) noise of the pump as well as calculated thermorefractive noise (green) are also plotted to constitute the technical noise in the measurement.

with Λ the duty cycle. The detailed derivation is provided in Supplementary Materials Section IC. Apparently, the responsivity has a periodic dependence on $n$. We note that, for quantum fluctuations coupled to modes with indices close to integer multiples of 1/Λ, they tend to cause breathing-type motions rather than timing jitter. This conclusion is validated by the numerically simulated responsivity plotted in Fig. 1d.

To verify the theoretical model, we perform experiments on AlGaAs-on-insulator (Al$_{0.2}$Ga$_{0.8}$AsOI) platforms, in which the very large Kerr nonlinearity ($n_2 = 1.7 \times 10^{-17}$ m$^2$ W$^{-1}$)[37–39] greatly enhances the coupling between DKSs and quantum fluctuations according to Eq. (2). The geometry of the microresonator is engineered to achieve normal dispersion in telecommunication C-band, with intrinsic quality factors exceeding 1.4 million as displayed in Fig. 2a. By launching 50 mW from a continuous-wave laser into the bus waveguide via a lensed fiber, dark pulses are generated through a deterministic process as facilitated by avoided mode crossings[10,17,18,20], featuring 100-nm-spanning optical spectrum and 91.47 GHz repetition frequency (Fig. 2b). Characterization of the timing jitter is based on a multi-spectral delayed self-heterodyne interferometer[31,40]. As shown in Fig. 2c, the amplified dark pulse is either frequency-shifted or temporally delayed in two split pathways, and the recombined signals are sent into a programmable optical filter. Two comb lines are selected for photodetection with their phases extracted by the Hilbert transform. According to the comb structure, their instantaneous phases are defined by the carrier-envelope offset frequencies, repetition frequencies, and mode indices. The timing jitter is computed by subtracting the common phases associated with the carrier-envelope offset frequencies (see Methods). The accuracy of such an interferometric approach is further validated using an electro-optic frequency comb, which is discussed in Methods.

In the experiment, the ±2nd comb lines relative to the pump are chosen. The measured jitter spectral density shown in Fig. 2d is close to the simulated quantum timing jitter at offset frequencies between 100 kHz and 1 MHz and is then limited by photodetection and digitizing noises at higher frequencies. Deliberate calibration is performed to eliminate noise from other origins. The noise of the pump laser

would induce timing jitter by shifting the spectral envelope center of the dark pulse via higher-order dispersion[12] or avoided-mode-crossings[41]. Note that the transduction from the phase noise of the pump laser to the timing jitter is also influenced by the photothermal effect at low offset frequencies. Therefore, we reveal this contribution by aligning the levels of calibration tones observed in both spectra of timing jitter and laser noise at multiple offset frequencies[42]. Specifically, the native intensity noise peak at 65 kHz and a series of artificial peaks generated by phase modulation is used. Timing jitter induced by the inherent thermorefractive noise of the comb-forming mode is also computed using a finite element solver[43]. Quantum decoherence apparently overrides the classical noise at offset frequency beyond 100 kHz.

In view of the $1/f^2$ dependence of its spectral density, quantum decoherence of dark pulses is a diffusion process, in which the variance of wavepacket locations would increase with time at a constant rate $D$. The diffusion coefficient $D$ introduced here can be utilized to evaluate the rates of quantum decoherence regardless of the FSRs of the devices. We examine the diffusion process by additionally injecting amplified spontaneous emission (ASE) noise into the bus waveguide (Fig. 3a). The ASE noise generated by an independent optical amplifier through spontaneous emission is deemed a semi-classical analogy of the quantum noise[30]. The timing jitter is observed to be lifted by increasing the ASE power, while the $1/f^2$-trend is maintained. The diffusion coefficients are shown to be linearly dependent on the input ASE noise power in Fig. 3b, and the total responsivity of timing jitter to the ASE noise power in the bus waveguide is derived from the fitted slope as 2.5 ps$^2$ s$^{-1}$ μW$^{-1}$. Therefore, the stochastic nature of the diffusion process is confirmed. We further characterize the spectral dependence of the responsivity by restricting the impact of the ASE noise to a single mode. It is realized by selecting 40-GHz-span of the ASE noise using the programmable filter. Line-by-line characterization of the responsivity is presented in Fig. 3c, which shows a periodic dependence on the relative mode number. This tendency is consistent with the mechanism predicted by Eq. (4), validating the SW model of quantum decoherence.

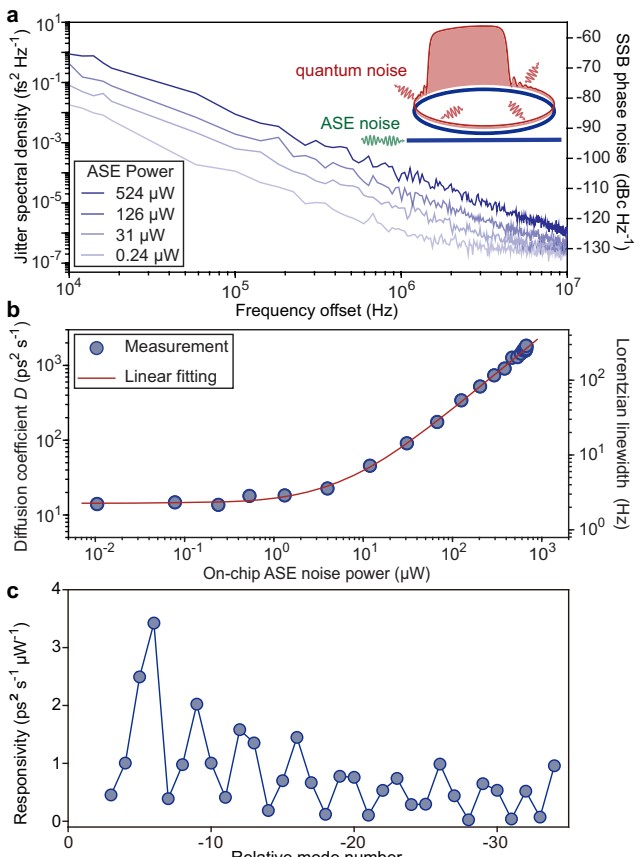

**Fig. 3 | Diffusion coefficient and responsivity. a** Jitter spectral density at different ASE power level. ASE noise generated from an EDFA is combined with the continuous-wave pump and sent into the microresonator. **b** Diffusion coefficients (Lorentzian linewidth of repetition frequencies) versus ASE power. The red line is a linear fit. **c** Measured jitter responsivity as a function of relative mode number.

The uncorrelated evolution of the spatially-separated SWs would deteriorate the uniformity of comb spacing. It is crucial to the practical implementation of optical-to-microwave conversion, but cannot be directly inferred from the radiofrequency beatnotes between the comb lines[44]. We revisit this issue for quantum-limited dark pulses by measuring the phase correlations between multiple comb lines. The protocol is described in Fig. 4a, in which the phase evolution of three comb lines is simultaneously recorded. Using one comb line as the common reference, two sets of phase evolution associated with the repetition frequencies are derived, whose correlation is expected to be unity if the comb is perfectly equidistant (see Methods). In the experiment, by reference to the 2nd comb line, we measure the phase correlations between 9 comb lines with relative mode numbers from −3 to −11, as shown in Fig. 4b. A bandpass filter spanning 100 kHz to 1 MHz is applied on the phase evolution to eliminate the impact of other noise sources. The correlations have a mean value of 0.9886 and a standard deviation of 0.0069, which is primarily limited by the uncorrelated digitizing and photodetection noises in the measurement. In light of the 2-Hz fundamental linewidth, we infer that the uniformity of comb spacing is better than 22.8 mHz. It can also be expressed in relative accuracy[44] as $1.2 \times 10^{-16}$ and $2.5 \times 10^{-13}$ when normalized to the optical carrier frequency and repetition frequency, respectively. Such a level of uniformity, which is verified by beating the dark pulse with an electro-optic frequency comb (see Methods), should be sufficient for a wide class of microwave photonic applications[45].

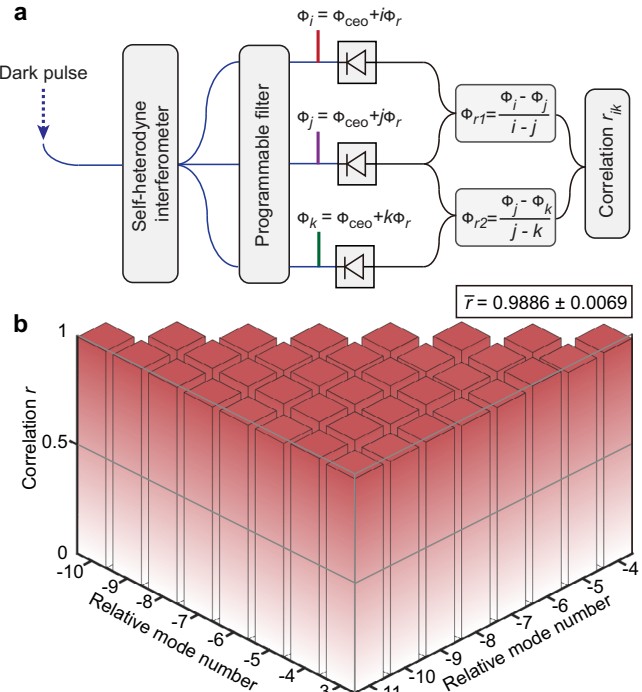

**Fig. 4 | Phase correlation of comb lines. a** Schematic illustration of phase correlation measurement. **b** Correlation map of −3rd to −11th comb lines.

## Discussion

The quantum decoherence of bright solitons and dark pulses are further compared in Fig. 5. Figure 5a showcases two DKSs simulated by identical parameters except for their opposite GVDs. The evolution of the pulse location in the rotating frame is computed 2000 times, and the results are overlaid in Fig. 5b. Statistics of the pulse locations at the end of the evolution imply that the dark pulse diffuses 4.5 times slower than its bright counterpart. The corresponding jitter spectral density of the dark pulse is 13 dB lower than that of the bright soliton (Fig. 5c). Such advantage of dark pulses, whose origin may have multiple facets, is also supported by numerical results using a broader selection of parameters in Supplementary Materials Section IE. We speculate that the higher power conversion efficiencies and more concentrated spectral profiles of the dark pulses benefit the coherence.

So far, several works have reported observations of quantum decoherence of DKSs. From Eqs. (1) and (2) we define a waveform factor $\alpha$ as

$$\alpha(\zeta, f, \text{sgn}(d_2)) = \frac{Dn_o A_{\text{eff}} D_1}{\sqrt{|d_2|n_2}}. \qquad (6)$$

Although the diffusion coefficient $D$ is related to pumping conditions and device parameters, $\alpha$ is solely dependent on normalized pumping conditions $(\zeta, f)$ and the sign of the dispersion $(\text{sgn}(d_2))$, which could serve as a common basis to compare the rates of quantum decoherence of different waveforms. Table 1 shows the results obtained in silica[30,31], Si$_3$N$_4$[32] and AlGaAsOI microresonators. While the derived $\alpha$ of bright solitons are larger than $1.92 \times 10^{-5}$ W, for dark pulse the $\alpha$ can be as small as $6.5 \times 10^{-6}$ W. It thus indicates that by choosing dark pulses the quantum-limited coherence could be improved by at least 5 dB in practical systems. Indeed, the 2-Hz Lorentzian linewidth of the 91-GHz-rate microwave signal here is already compelling among integrated optoelectronic oscillators[19,22,45,46]. Inferred from Eq. (6), migration to platforms with weaker Kerr nonlinearity should allow for coherence on a par with bench-top instruments.

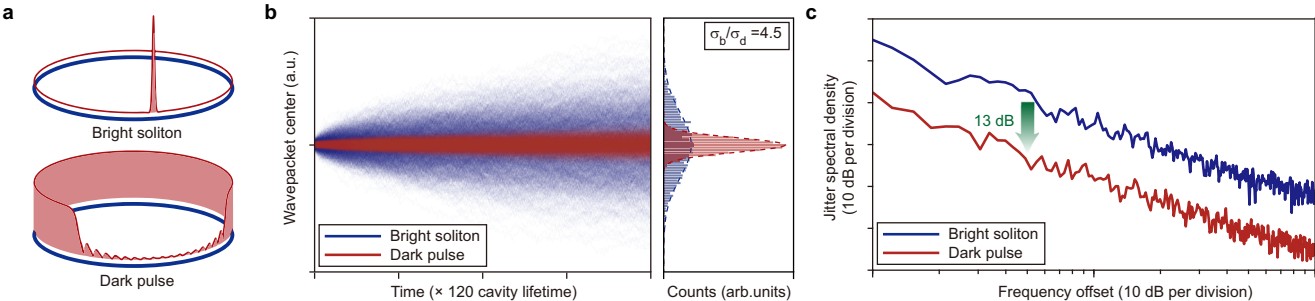

**Fig. 5 | Performance comparison between bright solitons and dark pulses. a** Intensity profiles of a bright soliton and a dark pulse. Parameters used in the simulations are: $f = 5.86, \zeta = 14, |d_2| = 0.0032$. **b** Left panel: 2000 simulated traces showing the evolution of the wavepacket center under quantum fluctuations. Right panel: histograms showing the distribution of the wavepacket center at the end of evolution. Inferred from the Gaussian fit (dashed lines), the standard deviation of the wavepacket center for the bright soliton ($\sigma_b$) is 4.5 times that of the dark pulse ($\sigma_d$). **c** Jitter spectral density derived from the traces in **b**. The offset frequencies are normalized to the microresonator linewidth and are plotted in arbitrary units.

### Table 1 | Experimental results on quantum decoherence of DKSs

| Material | DKS type | $n_o$ | $n_2(\mathrm{m^2W^{-1}})$ | $A_{eff}$ (µm²) | FSR (GHz) | $d_2$ | $D$ (ps²s⁻¹) | $\alpha$ (×10⁻⁵ W) | Ref # |
|---|---|---|---|---|---|---|---|---|---|
| Silica | Bright | 1.45 | $2.2 \times 10^{-20}$ | 18.3 | 21.9 | $6.2 \times 10^{-3}$ | $2 \times 10^{-2}$ | 4.22 | 30 |
| | Bright | 1.45 | $2.2 \times 10^{-20}$ | 33.0 | 9.4 | $4.7 \times 10^{-3}$ | $2.6 \times 10^{-2}$ | 4.87 | 30 |
| | Bright | 1.45 | $2.2 \times 10^{-20}$ | 44.1 | 22 | $8.5 \times 10^{-3}$ | $5.8 \times 10^{-3}$ | 2.53 | 31 |
| Si₃N₄ | Bright (Dispersion managed) | 2.0 | $2.2 \times 10^{-19}$ | 1.26 | 89 | $3.1 \times 10^{-4}$ | $5.3 \times 10^{-2}$ | 1.92 | 32 |
| AlGaAsOI | Dark | 3.3 | $1.7 \times 10^{-17}$ | 0.31 | 91.5 | $-4.1 \times 10^{-3}$ | 12.1 | 0.65 | This work |

Mean values of dispersion and mode area are used to calculate α for the Si₃N₄ microresonator reported in ref. [32].

Beyond the context of classical frequency combs, our discoveries have enriched the understanding of quantum dynamics in nonlinear optical systems. Simple access to the quantum-limited regimes of DKSs in AlGaAsOI microresonators is a pivotal step towards harnessing the quantum optical properties of microresonator frequency combs. We thus envisage wavelength-multiplexed quantum resources[5,47], multipartite entanglement[7] and quantum soliton phenomena[48] on these highly-nonlinear photonic chips.

## Methods

### Lugiato–Lefever equation and Langevin force
The Lugiato–Lefever equation in its full form reads

$$\frac{\partial A(\phi, T)}{\partial T} = i\frac{D_2}{2}\frac{\partial^2 A}{\partial \phi^2} + ig|A|^2 A - \left(\frac{\kappa}{2} + i\delta\omega\right)A + \sqrt{\frac{\kappa_{ext}P_{in}}{\hbar\omega_o}}. \quad (7)$$

Here $|A|^2$ is the photon density, $g = \frac{\hbar\omega_o^2 n_2 D_1}{2\pi n_o A_{eff}}$ is the nonlinear coupling coefficient, $\delta\omega$ is the laser-cavity detuning, $\kappa_{ext}$ is the coupling rate with the bus waveguide, and $P_{in}$ is the pump power. This equation is equivalent to Eq. (1) by introducing $\psi = \sqrt{2g/\kappa}A, \tau = \kappa T/2, d_2 = D_2/\kappa, \zeta = 2\delta\omega/\kappa$, and $f = \sqrt{8g\kappa_{ext}P_{in}/\kappa^3\hbar\omega_o}$.

We decompose $A$ into a set of discrete optical modes such that $\sum a_\mu e^{i\mu\phi} = A(\phi, T)$. The Langevin force applied on a single mode should satisfy the fluctuation-dissipation theorem, yielding

$$<\epsilon_\mu(T)\epsilon_\mu^*(T')> = \frac{\kappa}{2}\delta_{\mu\mu'}\delta(T - T'), \quad (8)$$

$$<\epsilon_\mu^*(T)\epsilon_{\mu'}(T')> = <\epsilon_\mu(T)\epsilon_{\mu'}(T')> = 0. \quad (9)$$

$\delta_{\mu\mu'}$ is the Kronecker delta. Equation (2) is derived from these spectral components through discrete Fourier transform

$$\epsilon(\phi, T) = \sum_\mu \epsilon_\mu(T)e^{i\mu\phi}, \quad (10)$$

which is further normalized following the above-mentioned guideline.

### Devices
The microresonator used in this work is fabricated on epitaxial AlGaAs bonded to an oxidized silicon wafer. Deep ultraviolet lithography, etching, surface passivation, and cladding are employed to define the structure of the microresonator. More details of the fabrication process are provided elsewhere[37,38]. The measured mode family dispersion ($D_2/2\pi$) is −1.63 MHz. For dark pulse generation, the temperature of the microresonator is set at 23.9 °C to introduce a red-shift of the pump mode by 2.3 GHz as a result of avoided mode crossings.

### Timing jitter characterization
The jitter spectral density of the dark pulse is derived from the power spectral density (PSD) of the relative phases of the measured beatnotes at the two photodetectors, given by

$$S_t(f) = \frac{PSD[\Phi_i - \Phi_j]}{(i - j)^2}\frac{1}{4D_1^2\sin^2\pi f\tau_d}. \quad (11)$$

$\tau_d$ is the time delay between the two arms of the interferometer. To avoid divergence at offset frequencies satisfying $\sin \pi f\tau_d = 0$, we introduce the cut-off frequency as $1/\tau_d$, beyond which only data points at offset frequencies of $(N+1/2)/\tau_d$ are plotted with $N$ a positive integer. The Lorentzian linewidth of the repetition frequency is inferred from the $1/f^2$ noise section of jitter spectral density as $2\pi f^2 D_1^2 S_t$, and the diffusion coefficient $D$ equals $4\pi^2 f^2 S_t/3$.

### Thermorefractive noise
The thermorefractive noise is simulated using a finite-element method based on the fluctuation-dissipation theorem[42,43]. Parameters used in the simulation are: (AlGaAs) material density of AlGaAs $5.008 \times 10^3$ kg m⁻³, heat capacity 346.4 J kg⁻¹ K⁻¹, thermal conductivity 22.52 W m⁻¹ K⁻¹, thermo-optic coefficient $6.798 \times 10^{-5}$ K⁻¹; (silica) material density

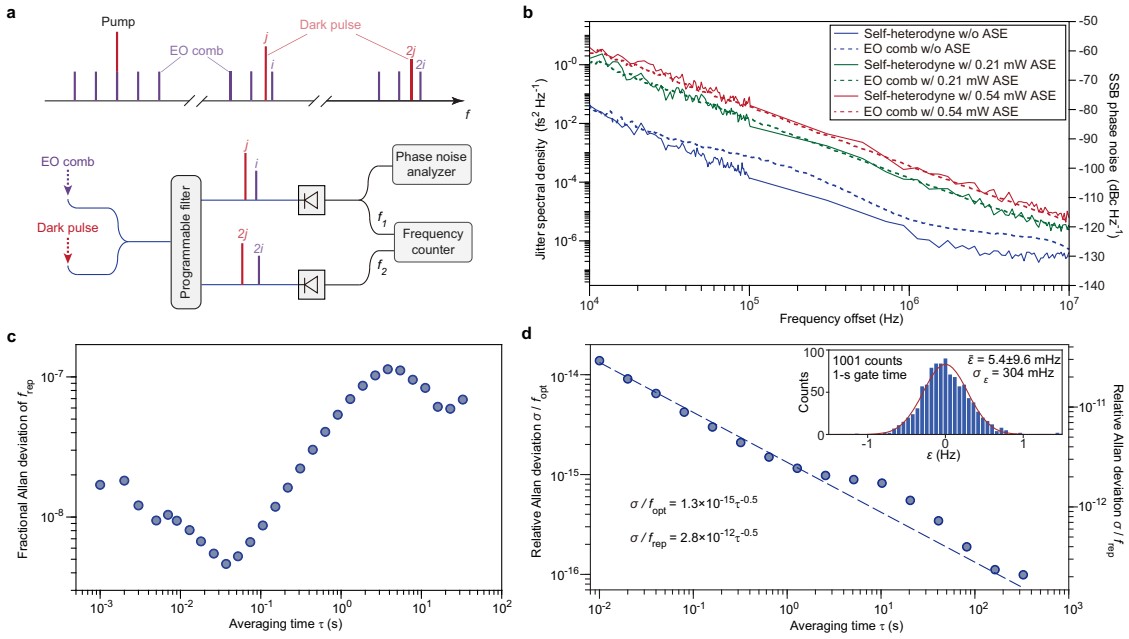

**Fig. 6 | Coherence of dark pulses measured by an electro-optic (EO) frequency comb. a** Schematic illustration of experimental setup. The EO comb is derived from the very same pump laser of the dark pulse using cascaded phase modulators and intensity modulators. Two pairs of comb lines are selected using a programmable filter for further characterization. **b** Timing jitter of the dark pulse with different injected ASE power. Measurements performed using the delayed self-heterodyne method and the EO comb are plotted as solid and dashed lines, respectively. **c** Fractional Allan deviation of the repetition frequency of the free-running dark pulse. The gate time is set to 1 ms. **d** Allan deviation of the deviation from equidistant mode spacing ($\epsilon$) normalized to optical carrier frequency ($f_{opt}$) and repetition frequency ($f_{rep}$). The gate time is set to 10 ms. The dashed line is fitted by the square root function given in the figure. Inset: distribution of 1001 points of $\epsilon$ measured at 1-s gate time. The red line is a Gaussian fit to the distribution.

---

$2.2 \times 10^3$ kg m$^{-3}$, heat capacity 703 J kg$^{-1}$ K$^{-1}$, thermal conductivity 1.38 W m$^{-1}$ K$^{-1}$, thermo-optic coefficient $1.2 \times 10^{-5}$ K$^{-1}$. During the simulation, ambient temperature (300 K) is assumed.

**Phase correlation**

The phases related to repetition frequencies are derived from two pairs of comb lines and are noted by $\Phi_{r1}$ and $\Phi_{r2}$. The correlation $r$ is given by

$$r = 1 - \frac{\int [\Phi_{r1}(t) - \Phi_{r2}(t)]^2 dt}{\sqrt{\int [\Phi_{r1}(t)]^2 dt \int [\Phi_{r2}(t)]^2 dt}}. \tag{12}$$

The uniformity of comb spacing expressed in relative accuracy is calculated as $(1 - r)f_L/f_{opt}$, in which $f_L$ is the Lorentzian linewidth of the repetition frequency ($\approx 2$ Hz) and $f_{opt}$ is the optical carrier frequency ($\approx 194$ THz).

**Measurement using an electro-optic (EO) frequency comb**

We construct an EO comb[49] to measure the coherence of the dark pulse so as to validate the results acquired by the delayed self-heterodyne method. The setup is described in Fig. 6a, where the EO comb ($f_{rep} = 25.4$ GHz) and the dark pulse microcomb ($f_{rep} = 91.5$ GHz) derived from the same pump laser are combined before sent into a programmable filter. The timing jitter of the beatnote between the $i_{th}$ comb line of the EO comb and the $j_{th}$ comb line of the dark pulse is primarily attributed to

$$S_{t,\text{beat}} = j^2 S_{t,DP} + i^2 S_{t,EO}. \tag{13}$$

Once the timing jitter of the EO comb is significantly smaller than that of the dark pulse, the timing jitter of the dark pulse can be inferred as $S_{t,\text{beat}}/j^2$. In the experiment, we choose $i = -18$ and $j = -5$ and the noise

data are recorded using a phase noise analyzer (Rohde & Schwarz FSWP50). The timing jitter of the dark pulse, which is controlled using excessive ASE noise, is plotted in Fig. 6b. The results inferred from the EO comb measurement agree well with those obtained using the self-heterodyne method for high-noise dark pulses. However, if the dark pulse is operated at its quantum limit, the timing jitter of the EO comb inheriting from the microwave driver (Keysight 8257D PSG Analog Signal Generator) becomes the limiting factor, which overrides the actual timing jitter of the dark pulse at frequency offsets beyond 40 kHz.

By recording the beatnote using a frequency counter, we derive the Allan deviation of the repetition frequency of the free-running, quantum-limited dark pulse. As presented in Fig. 6c, the fractional Allan deviation of the repetition frequency reaches $5 \times 10^{-9}$ at 40 ms averaging time. The rise of the Allan deviation at a longer averaging time is suspected due to the drift of the temperature, pump frequency, and coupling to the chip. We also characterize the uniformity of the comb lines by simultaneously comparing the beating frequencies of two pairs of comb lines[44] that are indexed by (i, j) and (2i, 2j). The deviation from equidistant mode spacing is defined as

$$\epsilon = \frac{f_2}{2j} - \frac{f_1}{j} = \left(1 - 2\frac{f_1}{f_2}\right)\frac{f_2}{2j}, \tag{14}$$

with $\frac{f_1}{f_2}$ given by the frequency ratio readings from the frequency counter. The Allan deviation of $\epsilon$ follows $\tau^{-0.5}$ trend with $\tau$ the averaging time. When normalized to the optical carrier frequency and the repetition frequency, the Allan deviation of the comb spacing uniformity respectively reaches $9.9 \times 10^{-17}$ and $2.1 \times 10^{-13}$ at averaging time of around 300 s. Furthermore, the distribution of $\epsilon$ acquired with 1-s gate time reveals the mean value of $\epsilon$ as $5.4 \pm 9.6$ mHz (inset of Fig. 6d), which is comparable with results reported in other literature[44].

## Data availability

The data that support the plot within this paper and other findings of this study are available at https://doi.org/10.5281/zenodo.7697733[50].

## Code availability

The codes that support the findings of this study are available at https://doi.org/10.5281/zenodo.7697733[50].

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

## Acknowledgements

The authors thank Prof. Chengying Bao at Tsinghua University and Prof. Wenjing Liu at Peking University for helpful discussions, as well as Yanwu Liu and Ze Wang for assistance in measurement. The project is supported by the High-performance Computing Platform of Peking University and the UCSB nano-fabrication facility.

## Author contributions

Experiments were conceived and designed by C.L., X.J., L.C., and Q.-F.Y. Measurements and data analysis were performed by C.L., X.J., and Q.-F.Y. Theoretical and numerical analysis was performed by X.J., H.W., Z.L., and Q.-F.Y. Devices were designed by H.S., L.C., W.X. and were fabricated by W.X. and L.C. The project was supervised by L.C., X.W., J.E.B., and Q.-F.Y. All authors participated in preparing the manuscript.

## Competing interests

The authors declare no competing interests.
