## [Peer Review File · Nature Communications]

Quantum decoherence of dark pulses in optical microresonatorsREVIEWER COMMENTS

Reviewer #1 (Remarks to the Author):

The manuscript, "Quantum decoherence of dark pulses in optical microresonators", by Chenghao Lao et al., reports on the measurement of quantum coherence and spectral density of phase noise of dark pulses with a repetition rate frequency of 91 GHz. The dark pulses are generated in a microresonator implemented on AlGaAs-on-insulator platform. I find the novelty fairly limited as dark pulses have been demonstrated 7 years ago by Gaeta's group. Additionally, I find that some of the claims in the paper are strongly overestimating the performance in terms of spectral purity of such dark pulses in microresonators. Some observations of course merits publication but I don't think they grant publication in a high impact journal.

1- The phase noise measurements have been realized with a fiber-delay based setup. These types of measurements entirely gloss over any frequency drift and random walk, inherent in an oscillator. I much prefer to see a phase noise measurement based on the experimental setup as described in Figure S6 of the supplementary material.

2- Coherence measurement cannot provide a stability or linewidth measurement. I read in the manuscript: "Nevertheless, in light of the 2-Hz fundamental linewidth, we infer that the uniformity of comb spacing expressed in relative accuracy⁴³ is better than 1.2×10^{-16} , meeting the criteria of miniature optical clocks²³ and a wide class of microwave photonic applications⁴⁴ ." I was, at first, very surprised by this performance and blown away. However, no explanations were provided in the main text, only in the method section this was made clear. Without reading the method section, the reader would be left in a state of utter confusion. Now, if the line by line spacing uniformity is under study, then the scaling needs to be done at the repetition rate frequency, i-e, ~91 GHz, which would then give, a linewidth of ~4kHz and a relative accuracy of 4.2×10^{-13} . Also, I repudiate that this meets the criteria of miniature optical clocks. Optical frequency combs are used to compare absolute optical frequencies or divide down an optical reference down to the microwave domain. None of the data presented here can back such claim.

3- Using the setup of Figure S6 with the 38 GHz phase modulator the authors should measure the Allan deviation to verify the true coherence of the line spacing for different averaging time.

In conclusion, there are some interesting observations (although not complete) in this manuscript, but I do not recommend publication in the present form, as the way data were taken lack a rigorous method to back some of the claims.

Reviewer #2 (Remarks to the Author):

The reviewed article is about noises in the normal dispersion microresonator frequency comb. Generation of frequency comb in normal dispersion regime is not so straightforward as in anomalous regime, but brings several benefits, thus the noise study is important for future applications. The article is well-written and also provides some theory on the solitonic pulse formation. The results look clear and justified. There are however several minor issues I want to point out.

1) It should be noted that solitonic pulses in normal dispersion are also called platicons [see 16,19,34,46 and others]. Essentially, dark pulse and platicon are different parts of the same entity, with platicon being the high-power and dark pulse -- the low power part of the waveform. Note also that, according to the LSW and RSW definition in fig 1c the formula for

the center after S25 represents the center of platicon, not dark pulse.

2) It is written "when both SWs are located in same regimes, they move in different directions so that the duty cycle breathes, causing intensity fluctuation of the dark pulse". Probably the authors meant energy or average intensity as the duration of the pulse, not the amplitude is said to change.

3) After formula (6) it is written that "alpha is solely dependent on normalized pumping conditions (ζ, f) and the sign of the dispersion ($\text{sgn}(d_2)$)", however in the formula (6) none of these variables are presented in the right side of the equality. Is it supposed that D has those dependencies? Then it probably should be written explicitly.

4) It would be good to see all the modeling parameters used to get fig S1a,b. It also would be illustrative to have the used pump parameter exactly under the Δ value in the picture.

5) What parameters were used for the modeling in fig. S2?

6) In the caption of figure S3 the detuning used for the modeling with AMX is not specified.

7) In the first supplementary section it was shown the equivalence of AMX and the complex change of the pump parameter. It looks like here the detuning was used to make solitonic pulses similar. Why was the methodology changed?

Finally I conclude that the paper should be accepted after addressing highlighted minor issues.

Reply to the reviewers for Nature Communications manuscript NCOMMS-22-39543-T.

We appreciate the careful review by the reviewers and have modified the manuscript in accordance with their suggestions. Here, we present a point-by-point reply (**in blue**) to the reviewers' comments (**in black**).

Response to the report from the Referee #1:

General comments: *“The manuscript, “Quantum decoherence of dark pulses in optical microresonators”, by Chenghao Lao et al., reports on the measurement of quantum coherence and spectral density of phase noise of dark pulses with a repetition rate frequency of 91 GHz. The dark pulses are generated in a microresonator implemented on AlGaAs-on-insulator platform. I find the novelty fairly limited as dark pulses have been demonstrated 7 years ago by Gaeta’s group. Additionally, I find that some of the claims in the paper are strongly overestimating the performance in terms of spectral purity of such dark pulses in microresonators. Some observations of course merits publication but I don’t think they grant publication in a high impact journal.*

In conclusion, there are some interesting observations (although not complete) in this manuscript, but I do not recommend publication in the present form, as the way data were taken lack a rigorous method to back some of the claims.”

Reply: We thank the reviewer for the general comments. Although dark pulses have been known to exist in microresonators since 2015, there have been no work reporting the fundamental noise of dark pulses so far. The main novelty of our work is the systematic study and the first experimental observation of quantum-fluctuation-induced decoherence of dark pulse microcombs. In the revised manuscript, we have provided additional experimental results to validate our claims. Details are discussed in “reply to comments 1-3”.

Comment 1: *“The phase noise measurements have been realized with a fiber-delay based setup. These types of measurements entirely gloss over any frequency drift and random walk, inherent in an oscillator. I much prefer to see a phase noise measurement based on the experimental setup as described in Figure S6 of the supplementary material.”*

Reply: We thank the reviewer for the comment. The multispectral delayed-self-heterodyne (MDSH) method has been adopted in a series of recent works to reveal the phase noise of optical frequency combs. In [K. Jung and J. Kim, Sci. Rep. 5, 16250 (2015).] and [D. Kwon et al, Sci. Rep. 7, 40917 (2017).], this method has been used to measure and suppress the timing jitter of fiber combs. In [Jeong, D. et al. Optica 7, 1108–1111 (2020).] and [Tetsumoto, T. et al. Nat.

Photon. 1–7 (2021)], the timing jitter of microcombs has also been resolved using MDSH interferometers. Besides showing a good agreement with other phase noise measurement technologies, the MDSH method has two advantages, namely the ability to measure high-rate combs and the high sensitivity in discriminating timing jitter. These features have made MDSH exceptionally suitable for measuring the timing jitter (phase noise) of microcombs with FSRs higher than the bandwidth of typical electronic instruments.

In the revised manuscript, we performed additional measurement to benchmark the performance of our MDSH setup. In the new Fig. 6 (also see below), we construct an electro-optic (EO) comb using the very same laser for pumping the microcomb. Photomixing a pair of comb lines of the microcomb and the EO comb produces radiofrequency beatnote, whose phase noise is analyzed using a commercial phase noise analyzer (Rohde & Schwarz FSWP50). The phase noise of the microcomb is translated from the beatnote noise and is compared with the results obtained from MDSH method. Microcombs with different noise levels are tested, which are controlled by varying the injected ASE noise. Excellent agreement between the two methods is apparent at high noise levels. However, for the microcomb at the quantum limit, the EO comb approach faces limit imposed by the phase noise of the microwave driver at offset frequencies over 40 kHz. Remarkably, the MDSH method shows improved sensitivity at this regime, based on which the actual quantum timing jitter is resolved.

Comment 2: *“Coherence measurement cannot provide a stability or linewidth measurement. I read in the manuscript: “Nevertheless, in light of the 2-Hz fundamental linewidth, we infer that the uniformity of comb spacing expressed in relative accuracy⁴³ is better than 1.2×10^{-16} , meeting the criteria of miniature optical clocks²³ and a wide class of microwave photonic applications⁴⁴.” I was, at first, very surprised by this performance and blown away. However, no explanations were provided in the main text, only in the method section this was made clear. Without reading the method section, the reader would be left in a state of utter confusion. Now, if the line by line spacing uniformity is under study, then the scaling needs to be done at the repetition rate frequency, i-e, ~ 91 GHz, which would then give, a linewidth of ~ 4 kHz and a relative accuracy of 4.2×10^{-13} . Also, I repudiate that this meets the criteria of miniature optical clocks. Optical frequency combs are used to compare absolute optical frequencies or divide down an optical reference down to the microwave domain. None of the data presented here can back such claim.”*

Reply: We thank the reviewer for the comment. Normalizing the uniformity level to the optical carrier frequency is a convention that has been adopted by [P. Del’Haye et al., Nature 450, 1214–1217 (2007)]. Nevertheless, we also agree that normalizing it to the repetition frequency would reflect the performance of the microcomb when they are used in optical frequency division. Therefore, in the revised manuscript, we added a sentence to clarify the relative accuracy:

“In light of the 2-Hz fundamental linewidth of the repetition frequency, we infer that the uniformity of comb spacing is better than 22.8 mHz. It can also be expressed in relative accuracy as 1.2×10^{-16} and 2.5×10^{-13} when normalized to the optical carrier frequency and repetition frequency, respectively.”

The claim that such uniformity meets the criterion for miniature optical clocks is now removed from the maintext.

Comment 3: *“Using the setup of Figure S6 with the 38 GHz phase modulator*

the authors should measure the Allan deviation to verify the true coherence of the line spacing for different averaging time.”

Reply: We thank the reviewer for the comment. In the revised manuscript, we have provided fractional Allan deviation of the repetition frequency of the free-running microcomb in Fig. 6c (also see below). Our result is comparable with those obtained in other integrated platforms such as Si₃N₄ (J. Liu et al., Nat. Photon. 14, 486-491 (2020)). The rise of Allan deviation at longer average time is attributed to drift of the temperature, pump frequency and the facet coupling, which are expected to improve once the device is packaged.

Moreover, we also provide the uniformity of comb spacing as a function of averaging time in Fig. 6d (also see below). The experimental scheme is similar to the setup described in [P. Del’Haye et al., Nature 450, 1214–1217 (2007)], where the beatnotes between two pairs of comb lines of the EO comb and the microcomb are simultaneously recorded in a frequency counter. The uniformity of the comb spacings are then derived from the ratio of the beating frequencies. We see that at 300 s averaging time the uniformity reaches 9.9×10^{-17} and 2.1×10^{-13} when normalized to the optical carrier frequency and the repetition frequency. Such level agrees well with our results obtained using the delayed-self-heterodyne interferometer.

Response to the report from the Referee #2:

General comments: *“The reviewed article is about noises in the normal dispersion microresonator frequency comb. Generation of frequency comb in normal dispersion regime is not so straightforward as in anomalous regime, but brings several benefits, thus the noise study is important for future applications. The article is well-written and also provides some theory on the solitonic pulse formation. The results look clear and justified. There are however several minor issues I want to point out.*

Finally I conclude that the paper should be accepted after addressing highlighted minor issues.”

Reply: Thanks for the reviewer’s constructive suggestions and comments. We made the point-by-point response as follows.

Comment 1: *“It should be noted that solitonic pulses in normal dispersion are also called platicons [see 16,19,34,46 and others]. Essentially, dark pulse and platicon are different parts of the same entity, with platicon being the high-power and dark pulse -- the low power part of the waveform. Note also that, according to the LSW and RSW definition in fig 1c the formula for the center after S25 represents the center of platicon, not dark pulse.”*

Reply: We thank the reviewer for pointing out this issue. In the revised manuscript, we have made the point clearly that the center refers to the **energy center** of the dark pulse and the duty cycle refers to the portion of the upper level. We also add a statement in Paragraph 2, Page 2 as

“Coexistence of the two disparate levels in the microresonator is accompanied by localized wave fronts known as the switching waves (SWs), which appear in pairs to form dark pulses along with the lower level. Note that the flat-top pulse comprising the upper level and the SWs are often referred to as platicons.”

Comment 2: *“It is written “when both SWs are located in same regimes, they move in different directions so that the duty cycle breathes, causing intensity fluctuation of the dark pulse”. Probably the authors meant energy or average intensity as the duration of the pulse, not the amplitude is said to change.”*

Reply: We thank the reviewer for pointing out this issue. In the revised manuscript, we have made changes in accordance with the comment above, which reads

*“when both SWs are located in same regimes, they move in different directions so that the duty cycle breathes and **the pulse energy fluctuates.**”*

Comment 3: *“After formula (6) it is written that “alpha is solely dependent on normalized pumping conditions (zeta, f) and the sign of the dispersion (sgn(d²))”, however in the formula (6) none of these variables are presented in the right side of the equality. Is it supposed that D has those dependencies? Then it probably should be written explicitly.”*

Reply: We thank the reviewer for the comment. Yes, the diffusion coefficient D has dependencies on ζ , f and $\text{sgn}(d^2)$. To clarify, in Page 5, right after Eq. 6, we have added the statement

“Although the diffusion coefficient D is dependent on both the pumping conditions and device parameters, α is solely dependent on normalized pumping conditions (ζ , f) and the sign of the dispersion ($\text{sgn}(d^2)$).”

Comments 4&5&6: *“It would be good to see all the modeling parameters used to get fig S1a,b. It also would be illustrative to have the used pump parameter exactly under the Δ value in the picture.”*

“What parameters were used for the modeling in fig. S2?”

“In the caption of figure S3 the detuning used for the modeling with AMX is not specified.”

Reply: We thank the reviewer for the comment. In the revised manuscript, we have provided all modeling parameters in the captions. Additionally, a detailed description of how to get dark pulses with different duty cycles in systems with and without AMX is added to the supplementary information text.

“In the systems with AMX, we change the detuning values ζ from 4 to 14 to get dark pulses with different duty cycles. Due to the degenerate characteristic of the dark pulses in the systems without AMX[6], we used square waves with different duty cycles to seed dark pulses with different duty cycles but with the same pumping parameters.”

Comment 7: *“In the first supplementary section it was shown the equivalence of AMX and the complex change of the pump parameter. It looks like here the detuning was used to make solitonic pulses similar. Why was the methodology changed?”*

Reply: We thank the reviewer for the comment. Although equivalence between systems w/ and w/o AMX is established in Fig. S1, it requires precise tuning of all pumping parameters. Since the purpose of Fig. S3 is to showcase the dependence of the modulation period of the responsivity on the duty cycle, it is not necessary to test on identical dark pulses.

REVIEWERS' COMMENTS

Reviewer #1 (Remarks to the Author):

I thank the authors for their answers and additional experiments. I appreciate that the Allan deviation is now shown and reveals some actual frequency drifts. I agree that fiber-delay-line based measurement can enhance the sensitivity when measuring the phase noise and it is better when shown with an Allan deviation. I now believe the manuscript is suitable for publication in Nature communication.

Reviewer #2 (Remarks to the Author):

The authors addressed all issues that were pointed out previous time and I think the article can now be published.